# Attempting AG-Doped Diamond-Like Carbon Film to Improve Seal Performance of Hydraulic Servo-Actuator

**DOI:** 10.3390/ma13112618

**Published:** 2020-06-08

**Authors:** Zhiyan Zhao, Xiang Yu, Zhiqiang Zhang, Wen Shu, Jia Li

**Affiliations:** 1School of Engineering and Technology, China University of Geosciences, Beijing 100083, China; 3002150016@cugb.edu.cn (Z.Z.); 2102130021@cugb.edu.cn (W.S.); 2Beijing Key Laboratory of Materials Utilization of Nonmetallic Minerals and Solid Wastes, National Laboratory of Mineral Materials, School of Materials Science and Technology, China University of Geosciences, Beijing 100083, China; 3003180023@cugb.edu.cn (Z.Z.); 2103160025@cugb.edu.cn (J.L.)

**Keywords:** diamond-like carbon film, Ag doping, seal performance, hydraulic servo-actuator

## Abstract

A hydraulic servo-actuator is a critical aircraft control device whose sealing performance directly affects the sensitivity and accuracy of the aircraft flight attitude. Foreign intrusive particles in hydraulic oil may induce the vulnerable friction pair wear and the seal leak; they may even lead to oil spill accidents. This work attempts to conduct a systematical investigation of Ag-doped diamond-like carbon (Ag-DLC) film to improve the seal performance. The failure of the servo-actuator was analyzed. Then, a series of Ag-DLC films was deposited; the structure and combined tribological performances of the Ag-DLC films were investigated. The results show that the intensity of the Ag (111) crystal face in the films increases with an increase of Ag content. The hardness, intrinsic stress, frictional coefficient, and wear rate of the films tend to decrease with the amount of doping metal. The a:C-Ag_10.5%_ film exhibits optimal combined properties. The Ag doping makes the film toughness improve; both soft Ag particles and a graphitized top layer act as solid lubricants. Our findings may offer a novel approach to make DLC film applicable for improving the seal performance of hydraulic servo-actuator. Based on the experimental data, a mechanism behind the film modification of Ag-DLC film is also revealed.

## 1. Introduction

A hydraulic servo-actuator is a critical aircraft control device whose sealing performance directly affects the sensitivity and accuracy of the aircraft flight attitude [1,2]. Foreign intrusive particles in hydraulic oil may induce the vulnerable friction pair wear and the seal leak, and they may even lead to oil spill accidents [3,4]. Film modification thereon is an advisable approach to improve the tribological performances for the friction pair [2,5]. W and Cr containing diamond-like carbon (DLC) are two commonly used films, but such carbide metal doping suffers from early failure due to a poor toughness [6,7]. None carbide-forming metals such as Ag may bring hope to replace their carbide-forming counterpart. Our previous Ag-DLC research [8] indicates that a soft and ductile Ag, embedded into the amorphous carbon matrix, may improve the tribological properties by lowering brittleness rates. Unfortunately, a systematic investigation is rare to see in the literature for using Ag-DLC film to improve the seal performance of a hydraulic servo-actuator.

Such systematic investigation should contain three aspects: (1) the leak source and film service demand for the friction pair; (2) the film preparation and verification; and (3) the film modification mechanism. Two available methods for the seal failure are as follows. (1) Optimize the design for the friction pair to reduce the friction between the components, and (2) use heat treatment for materials to improve the component performances. The available methods cannot meet the service requirements and are suspected for lack of three important data [9]. (1) Identify the part to be worn, and figure out the service demands for the film through failure analysis. (2) Deposit Ag-DLC film to meet the service demands, and verify the film feasibility by structural characterization and wear test. (3) Explore the modification mechanism of the film applicable for the activator. Only such systematical investigation can make a suitable Ag-DLC film happen.

Therefore, this work attempts to conduct a systematical investigation of Ag-doped diamond-like carbon film to improve the seal performance of a hydraulic servo-actuator. The failure of the servo-actuator is analyzed, and the film service demand is suggested. A series of Ag-DLC films are deposited on an AISI 440 piston rod, and the combined tribological performances of the Ag-DLC films are investigated. Consequently, the film modification how Ag-DLC film improves the seal performance is revealed.

## 2. Seal Failure Analysis and Film Modification

### 2.1. Structural Composition and Working Principle

Figure 1 exhibits a physical map and schematic diagram of the hydraulic servo-actuator. With regard to the structural composition shown in Figure 1a, the main role of the actuator lies in transforming the point command signal of the aircraft control system into an electro-hydraulic signal with a certain level of power, such that the speed and attitude of the aircraft can be controlled. The control circuit is completed as follows. The actuator cylinder can be denoted as a hydraulic ram or hydraulic cylinder block, and it consists of cylinder 1 and piston 2. The piston rod 3 drives the actuator to move through the exertion of pressure from the actuator cylinder [10]. The piston rod 3 and the seal ring 5 on the seal structure rub against each other, and the signal is subsequently transmitted to a signal feedback device; the feedback device converts these signals into the corresponding electrical signals by detecting the changes in the displacement or the velocity of piston rod 3.

Figure 1b shows the working principle of the actuator. The actuator overcomes the pressure load by promoting the oil flow. Thus, the speed and frequency of the reciprocating movement of the piston rod 3 and the cylinder 1 depends on both the flow rate and velocity of the oil. The signal is subsequently transmitted to the signal feedback device 7, and it controls the operation of the hydraulic system. During operation, the right part of the cylinder in the actuator remains constant. The oil pressure in the left cylinder is increased as the left side of the inlet is fed. Once the oil pressure reaches a normal value, piston 2 is pushed to move to the right by the hydraulic oil. Piston rod 3, which is connected to the piston, is driven to the right. Oil is supplied to the interior continuously as the oil passes through the inlet. Thus, the piston provides a continuous reciprocating movement at a certain speed.

### 2.2. Seal Failure Process

As mentioned above, the seal failure process can be divided into three stages.

Stage I. During the piston rod returns, the scraping ash board in the front of the piston rod may hinder large-sized particles from leaking into the internal system. However, small-sized particles still may leak into the internal system or blend in the gap between the friction pairs dragged by the hydraulic oil.

Stage II. Solid particles, attached to the surface of the piston rod, are embedded in the seal during the reciprocation step. The seal ring is composed of a soft plastic retainer and a ball bearing. Since metallic debris (i.e., oil-suspended solid particles) possesses higher strength and hardness than the plastic, the solid particles are readily embedded in the retainer under the high internal pressure inside the actuator.

Stage III. During the reciprocating motion of the piston rod, the contaminants embedded in the surface of the seal ring can act as abrasives, rubbing against the piston rod. This causes abrasive wear on the surface of the piston rod, and it eventually results in the formation of a wear scar. The abrasive wear worsens with time and may lead to oil leakage, and, ultimately, to an actuator seal failure.

### 2.3. Seal Failure Analysis and Film Modification Approach

The piston rod follows a reciprocating motion during operation. The average linear velocity ranges from 4.5 to 5.0 m/s; the peak stress, the environment temperature, and the relative humidity are in the ranges of 300–400 MPa, 80–160 °C, and 80–85%, respectively [11]. During the continuous reciprocating motion within the cylinder, the piston rod has to overcome two kinds of loads namely, the interactive change of tension and pressure and the impact load from the feedback device. When the actuator starts, the piston rod follows a reciprocating motion after the feedback device sends a signal. In case that the piston rod suddenly changes directions during the reciprocating motion, the solid contaminants and the piston rod rub against each other. As a result, the surface of the piston rod has to overcome a larger contact stress (as high as 10 GPa), and this can induce wear and deteriorate their seal performance.

The hardness of these contaminant particles has a substantial effect on the system wear. A positive correlation between the hardness of the contaminants and the surface wear is previously proposed [12]. According to this correlation, the material may undergo surface wear to a low extent when the contaminant particles are lower or equal in hardness compared to the surface material [13]. The metal surface may undergo wear when the hardness of the contaminant particles is larger than that of the metal surface. Wear can be considered negligible when the surface hardness of the piston rod is significantly higher than that of the particles.

During actuator operation, the piston rod and the seal undergo a high-speed reciprocating motion, and the extended end of the piston rod is prone to undergo abrasive wear. The failure rate of the sealing device can be reduced by increasing the wear resistance of the piston rod. The sectional diameter of the seals is 2.6 mm, while the fixed width of the seal ring groove is 3.5 mm. Once the actuator is installed, the initial gap between the piston rod and the seal ring is 0.6 mm. As a result, the deposition of Ag-DLC films on the piston rod [13,14] is expected to enhance its abrasion resistance, thereby improving the wear resistance of the frictional pair without affecting the initial fit clearance of the piston rod [14,15].

## 3. Experimental Details

### 3.1. Sample Pretreatment

The substrate was two wafers; silicon (commercial, Ra (surface roughness) less than 0.1 μm) (Beijing Chemical Workstation, Beijing, China) was used for structural and mechanical analyses, and AISI 440 (commercial raw material of piston rod, Ra (surface roughness) less than 0.2 μm) (Beijing Chemical Workstation, Beijing, China) was used for tribological analysis. The substrate was cleaned with acetone in an ultrasonic bath for 20 min. After drying with clean nitrogen, the substrate was placed in a vacuum chamber ready for deposition.

### 3.2. Film Deposition

Figure 2 shows a schematic diagram of multi-ion beam assisted deposition (IBAD) system (SP9060, PowerTech, Beijing, China). The base vacuum was set to 1.8 × 10^−4^ Pa, and the deposition pressure was fixed at 1.5 × 10^−2^ Pa. The substrate was initially bombarded with an Ar^+^ beam of 15 kV/20 mA (ion source voltage/ion beam current) for 10 min for a fresh and clean surface. Subsequently, a 0.2 μm-thick Ag interlayer was deposited by singular sputtering of Ag target (silver, purity of 99.99%) (Zhongnuo Advanced Material Technology Co., Ltd., Beijing, China) at 1200 eV/35 mA. Then, a 0.8 μm-thick layer (a:C-Ag_x_) was synthesized by the simultaneous co-sputtering of Ag and C targets (carbon, purity of 99.99%) (Zhongnuo Advanced Material Technology Co., Ltd., Beijing, China). The co-sputtering was performed by (1) the Ag sputtering current varying from 0 to 50 mA while maintaining the energy constant at 800 eV; and (2) constant C sputtering current and energy of 80 mA and 1300 eV, respectively. Six samples of Ag-DLC films (approximately 1 μm-thick) were prepared and denoted as A0–A5 in Table 1.

### 3.3. Film Structure Analysis

The elemental composition was detected using energy-dispersive spectroscope (EDS, JSM 6301F, JEOL, Tokyo, Japan). The Ag grain size was evaluated using X-ray diffraction (XRD, Model XD-3, Rigaku, Tokyo, Japan). The bonding structure was analyzed using Raman spectroscopy (LabRAM HR Evolution, HORIBA, Kyoto, Japan).

### 3.4. Mechanical Test

The hardness was measured on a nanoindenter (MTSXP, MTS, Minneapolis, MN, USA) under a load of 2 mN, in a continuous stiffness measurement test pattern. The intrinsic stress was calculated using the Stoney formula [15,16], and the thickness of the silicon substrate and the film was measured with a three-dimensional white-light profiler (ManoMap-D, AEP, Columbus, OH, USA).

### 3.5. Tribological Test

Conforming to the actual working conditions, the tribological performance of the films was evaluated via reciprocating friction on a ring-on-block tester (MFT 4000, Huahui, Lanzhou, China) under the following conditions. (1) The coated sample was fixed, and a plastic ring was placed next to the sample surface. (2) A reciprocating motion was conducted on the sample surface at a load of 5 N. (3) An optional oil box containing Poly Alpha Olefin4 (PAO 4) was employed, and the base-oil applied in this test had a kinematic viscosity of 3.86 mm^2^/s at 100 °C. The application of additive-free oil permitted the evaluation of the performance on the friction and wear of the DLC film itself devoid of any impact from the additive in lubricant. (4) The temperature and the relative humidity were, respectively, 100 °C and 80% in a wet friction test. (5) The amplitude, the reciprocating frequency, and the duration was 30 mm, 10 Hz, and 15–60 min, respectively. (6) The wear morphology was observed on an optical microscope (OM, BX51M, OLYMPUS, Tokyo, Japan) provided with an attached digital camera. To ensure reliability, the obtained values for each sample were the average of six measurements, and the accuracy was within a ±5% error range.

## 4. Results and Discussion

In Section 2, the actuator failure is analyzed, and the film modification approach is then suggested. In this part, we discuss the influence of the Ag doping content on the structure and on their mechanical and tribological performances of the films. Consequently, we propose a mechanism of how Ag-DLC film improves the seal performance.

### 4.1. XRD Analysis of Ag-DLC Films

Figure 3 shows XRD patterns of Ag-DLC films with five contents. As shown in Figure 3, the crystallographic peak corresponding to Ag (111) broadens while decreasing content of the Ag doping. This peak broadening may be associated with the presence of grains with small size [17]. Both the Ag crystalline size and the relative intensity of the Ag (111) peak increase with the increasing Ag content. The slight and random shift of the Ag (111) peak around 2θ = 38.2° with the Ag content suggests changes in the size of the Ag crystals, which in turn results in a change of the film performances. The DLC film is typically amorphous, and the Ag crystallites are dispersed in this amorphous matrix [18]. Thus, the microstructure of the Ag-DLC films consists of a mixture of amorphous carbon and crystalline Ag phases. The Ag crystallites dispersed in the amorphous carbon network may create favorable sites for re-nucleating the Ag particles upon increasing the Ag content.

### 4.2. Hardness and Intrinsic Stress of Ag-DLC Films

Figure 4 shows the measured hardness and intrinsic (compressive) stress of the Ag-DLC films as a function of the Ag doping. As shown in Figure 4a, the hardness of the Ag-DLC films ranges within 13.6–23.6 GPa, and the value decreases with the increase Ag content due to the soft nature of the doped Ag. The hardness of the films is higher than the maximum working stress (10 GPa) of the system, and it is significantly higher than that of the AISI 440 substrate (ca. 7 GPa). Such film may enable the component thereon to have an outstanding wear resistance against the particles. Similarly, the compressive stress (Figure 4b) of the Ag-DLC films tends to decrease with the Ag content.

As an exception to the basic tendency is shown in Figure 4: the a:C-Ag_10.5%_ sample (doping silver at 10.5%) showed a significant hardness increase and a simultaneous decrease of the intrinsic stress. The dependence of the hardness and the intrinsic stress with the Ag content is rather complex, with two competing and contradicting factors being present in the Ag-DLC films. On one hand, the XRD analyses revealed that the Ag crystallites dispersed within the amorphous matrix may induce changes in the properties of the films, and the doping of Ag at a certain content and size may contribute to this abrupt change. On the other hand, the presence of soft and flexible Ag crystallites embedded in the amorphous carbon matrix may release strains and make the film stress reduce. Overall, the hardness and intrinsic stress of the films depend on the combined action of these two factors. An optimum content of Ag doped in an amorphous carbon matrix with a suitable crystallite size is likely to optimize the mechanical properties of the films [8], providing them with high hardness and low intrinsic stress, and in turn results in good toughness properties. This improved toughness may prevent seal leakage issues caused by the high oil pressure on the coated piston rod. The film hardness is significantly higher than that of the oil pollutant particles (i.e., solid particles mostly composed of steel debris). The lower intrinsic stress of the Ag-DLC films may prevent the film detachment under the serving conditions. Such a coated component may effectively reduce the probability of surface scratches when rubbing with the solid particles.

### 4.3. Tribological Performances of Ag-DLC Films

The damage produced by contaminants in the sealing structure induces the seal failure of the hydraulic servo-actuator [3]. This damage typically occurs between the seal ring and the piston rod. There is a thin oil film on the outer surface of the piston rod. During the reciprocating motion, the outer surface may attach some particulate contaminants, which are in contact with the film. The contaminants adhered to the rod surface can be transferred to the ring and aggregate. As a result, abrasive wear between the rod surface and the particles embedded in the ring takes place, resulting in the seal failure.

Figure 5 shows the steady-state coefficient of friction (COF) and wear rate of the Ag-DLC films after the reciprocating test. As shown in Figure 5a, the steady-state COF ranges from 0.05 to 0.17 with the Ag content. The Ag-DLC films show a lower COF than the undoped DLC (0.17), and a:C-Ag_10.5%_ (A2) shows the lowest value (0.05). The COF values of the Ag-DLC film samples remain below 0.2 in all cases, revealing an effective lubricating effect [19].

As shown in Figure 5b, the wear rate decreases with the Ag loading and increases after Ag content of 10.5%. The a:C-Ag_10.5%_ film exhibits the lowest wear rate (ca. 3.8 × 10^−9^ mm^3^/mN) among the films in this experiment. Comparing with the optimal Ag-DLC film prepared by M. Hua et al [8], and under the same load and sliding distance, the wear rate obtained herein was ca. 16 times lower than that reported by these authors (6.48 × 10^−8^ mm^3^/mN). It can be deduced from these results that the Ag-DLC films can prolong the life of the piston. Being a soft metal, the Ag particles embedded within the amorphous carbon network matrix can reduce brittleness. Additionally, Ag can provide a buffer space allowing stress concentration in the carbon matrix, thereby improving its tribological performances. As a result, the a:C-Ag_10.5%_ film shows optimal tribological properties, and it is further investigated below.

Figure 6 shows the morphologies of the wear scars obtained from sample A2 (a:C-Ag_10.5%_ film) during four frictional periods. As compared in Figure 6a and Figure 6b, the grinding marks and wear debris of the a:C-Ag_10.5%_ film gradually increase with the sliding time. The presence of debris particles accumulated on the worn edges implies the formation of a transfer layer. As shown in Figure 6c, as the film wear intensifies, furrow can be observed in the wear scar. Thus, some accumulated materials and breakages are observed within a fringe of the track, and the transfer layer becomes apparent. Figure 6d shows the wear scar of the counterpart represented in Figure 6c. The wear rate of the counterpart was higher than that of the Ag-DLC film, probably because the hardness of counterpart is significantly less than that of the film.

EDS analyses on the wear debris obtained from the friction pair indicated the presence of Fe, Cr, C, and Ag. Fe and Cr elements originated from the AISI 440 substrate, while C and Ag were derived from the DLC film. This result confirmed that the transfer layer resulted from the wear between the Ag-DLC film and the AISI 440 substrate. The Ag-containing transfer layer formed on the interface between the counterpart and the film was responsible for the low COF.

### 4.4. A Comparative Analysis of Raman Spectra of a:C-Ag_10.5%_ Film

Figure 7 compares the Raman spectra of undoped DLC film and a:C-Ag_10.5%_ film under a load of 5 N. In Figure 7, the four Raman spectra correspond to undoped DLC film, as-deposited films (no wear) before the frictional test, wear track, and debris of the films after the frictional test, respectively. In the Raman spectra, two structural aspects (i.e., G- and D-peaks and I_G_/I_D_ values) are individually discussed.

The Raman spectrum of the undoped DLC film exhibits two D and G characteristic peaks, at ca. 1395 and 1500 cm^−1^, respectively. As shown in Figure 7a,b, the G peak shifts from 1500 cm^−1^ (undoped DLC film) to 1550 cm^−1^ (no wear film), and the D peak shifts from 1395 cm^−1^ (undoped DLC film) to 1380 cm^−1^ (no wear film). This shift of the D peak moves toward a lower wavenumber while the G peak moves toward a higher wavenumber, which may result from the doping of Ag in the DLC film [8,20]. Consequently, the I_G_/I_D_ ratio increases from 1.4 (undoped DLC film) to 1.5 (no wear film), and the content of sp^3^-C decreases from 41.7% for undoped DLC film to 35.8% for no wear film. Such a change of carbon bonding structure may affect the film performances accordingly.

The Raman spectrum of the a:C-Ag_10.5%_ film exhibits two D and G characteristic peaks [21], at ca. 1380 and 1550 cm^−1^, respectively. As shown in Figure 7b–d, the G peak 1550 cm^−1^ (no wear film) shifts after the frictional test (1585 cm^−1^, debris, and 1590 cm^−1^, wear track). This shift to a higher wavenumbers implies an increase in the number of sp^2^-C bonds. Similarly, the D peak shifts from 1380 cm^−1^ (no wear film) to 1310 cm^−1^ (wear track). This shift of the D peak to lower wavenumbers is indicative of a lower number of sp^3^-C bonds. The heat generated during the reciprocating friction at the interface between the film surface and counterpart may let the temperature of the contact area rise. This may make the number of metastable sp^3^-C bonds decrease and the number of steady state sp^2^-C bonds increase.

As for the I_G_/I_D_ ratio (Figure 7b–d), it increases from 1.5 (no wear film) to 2.0 (wear track) and 2.2 (debris), where the content of sp^3^-C decreases from 35.8% for no wear film to 29.8% for wear track and to 27.9% for debris. This variation of the I_G_/I_D_ ratio may reflect a change of the film microstructure, and results from the tradeoff of two processes. On one hand, the structure and distribution varies with the number of sp^2^-C and sp^3^-C bonds. During the reciprocation test, the friction heat induces some sp^3^-C bonds to be converted into sp^2^-C bonds in the Ag-DLC films [22]. The generated sp^2^-C bonds may move to the film surface under the action of a contact force, and the corresponding drop in density of the carbon particles in local area induces the conversion of sp^3^-C into sp^2^-C bonds to adjust to the new conditions. On the other hand, Ag doping may alter the C–C covalent bond angle and the distribution of carbon particles. According to the molecular orbital hybridization theory [23], the atomic orbitals start to overlap to form molecular orbitals (MO) as atoms get closer. Ag has six filled d-orbitals, and the orbitals overlap with those of carbon, increasing the bond angle and minimizing the bond energy make the local area stable [24]. During the reciprocation test, the non-carbide forming Ag in the film undergoes plastic deformation and diffusion into the high-temperature region, inducing the silver to distribute in the film uniformly. In this case, Ag may empty more d-orbits, transforming sp^3^-C bonds into easily combined sp^2^-C hybrid bonds.

As shown in Figure 7, under the action of the frictional heat and Ag, the D-peak shifts to lower wavenumbers, revealing a reduction in the number of thesp^3^-C bonds. Likewise, the shift in G-peak position to higher wavenumbers indicates an increase in the number of sp^2^-C bonds. The increase of the I_G_/I_D_ ratio reveals the transformation from sp^3^-C to sp^2^-C bonds. The sp^2^-C bonds contribute to a favorable solid lubrication. Additionally, an increase in the number of sp^2^ bonds at the contact interface may result in films with low COF values. Thus, doping by soft and flexible Ag can release micro-strain in amorphous carbon matrixes via plastic deformation, thereby improving film toughness. These low COF and good toughness values result in films with good wear resistance properties. Thus, the tribological properties of the coated piston rod can be improved by doping appropriate amounts of Ag particles.

### 4.5. Mechanism Allowing Ag-DLC Film to Improve the Sealing Performance

This work is initiated to explore a Ag-DLC film applicable for improving the seal performance of a hydraulic servo-actuator. On the basis of the experimental data mentioned above, a mechanism allowing Ag-DLC film to improve the sealing performance is proposed.

Figure 8 illustrates the mechanism by which the Ag-DLC film improves the sealing performance between the piston rod and the seal ring of the actuator. Figure 8 contains four sections. Figure 8a is the front view of the friction pair between the coated piston rod and the solid particles (black block) embedded within the seal ring. Figure 8b shows a side view of the friction pair represented in Figure 8a, and the up–down double arrow indicates the reciprocating motion. Figure 8c shows a micrograph of the contact interface of the friction pair, and it includes the following layers from bottom to top: AISI 440 substrate, Ag-DLC film, graphite layer (black oval), transfer layer, and solid particles. Figure 8d shows a micrograph of the transfer layer represented in Figure 8c, and it includes the following layers from bottom to top: Ag-DLC film; transfer layer and solid particles; transfer layer comprising silver particles (silver ball), sp^3^-C (gold ball), sp^2^-C (blue ball), chromium atom (green ball), and the iron atom (brown ball).

The mechanism behind the improvement of the sealing performance via Ag-DLC film may be a compromise of three effects below.

(1)Contact force and friction heat induce graphitization. As shown in Figure 8c, the friction heat (S shape) during reciprocating motion results in the transformation of sp^3^-C bonds into sp^2^-C bonds in the Ag-DLC film. Under contact force and thermal diffusion, the sp^2^-C bonds generated diffuse to the surface of the Ag-DLC film and aggregate to form a graphitized top layer (black oval) on the film. Meanwhile, under a friction heat and contact force, a transfer layer is formed by the abrasive debris produced upon contact between the Ag-DLC film and the solid particles.The friction heat between the piston rod and the solid particles increases the temperature of the contact zone. Accumulation of the friction heat may lead to the occurrence of a flash temperature at the local micro-contact area. Consequently, the sp^3^-C bonds in high-heat metastable state are converted into sp^2^-C bonds in a high-heat steady state, which is in line with the Raman results shown in Figure 7. The generated sp^2^-C bonds diffuse to the top surface of the film under the contact force and friction heat. During the reciprocating movement, the increase in the frequency of the flash temperature may lead to a larger number of sp^2^-C bonds. Under the action of a contact force, the sp^2^-C bonds aggregate in the contact area and form a graphitized top layer (Figure 8c), and the layer plays an important role in lubricating the solids and reducing the COF of the friction pair.(2)Silver particles improve film toughness and act as a solid lubricant. As shown in Figure 8c and d, under the action of a contact force (blue arrow) in the reciprocating friction test, the Ag particles (white particles) present in both the Ag-DLC film and the transfer layer undergo plastic deformation (deformed shape). Subsequently, under friction heat and a contact force, the Ag particles diffuse and accumulate on the surface of both the film and the transfer layers, acting as a solid lubricant.The presence of doped Ag results in films with enhanced toughness as a solid lubricant. When the film is in contact with the particles embedded in the retainer, the contact force causes microstrain in the local area of the film, and the internal stress in the local area increases accordingly. Under the action of an internal stress, the Ag particles doped in the film undergo plastic deformation (Figure 8c,d). Such plastic deformation may release the surrounding microstrain, reducing the internal stress, which is in line with the intrinsic stress results (Figure 4), and showing a significantly lower stress upon increasing the Ag content. In this case, the toughness of the films can be improved, such that microcrack initiation/propagation and the peeling-off issue can be retarded. Under the co-action of the contact force and friction heat, the Ag particles undergo plastic deformation, diffusing and aggregating at the top surface of the film. These Ag particles having low shear stress act as solid lubricants. The improved toughness and solid lubrication characteristic upon the addition of Ag particles contribute to the improvement of the tribological properties of the films.(3)The transfer layer plays a self-lubrication role and extends the seal life. As shown in Figure 8c, the transfer layer is formed between the film and the solid particles under the action of a contact force and friction heat. This layer is formed to avoid the direct contact between the two frictional surfaces. In Figure 8d, Ag is deformed in the transfer layer and diffuses to the surface of the transfer layer.In the initial stage of the reciprocating friction, the contact force brings the two contact surfaces to wear, producing debris containing Ag particles, sp^3^-C, sp^2^-C, Fe, and Cr between the contact surfaces. Owing to the accumulation of the friction heat, sp^3^-C bonds are converted into sp^2^-C bonds and diffuse toward the surface of the film, resulting in the graphitization of the film surface (Figure 7). Under a contact force and friction heat, the graphite is generated between the two contact surfaces and reacts with the debris, forming a transfer layer. This transfer layer is formed to prevent direct contact between the solid particles and the film. When a force is applied on the transfer layer, solid particles are squeezed into the transfer layer, generating a high strain in local areas to increase the corresponding internal stress. Under the action of an internal stress, soft and flexible Ag undergoes plastic deformation, upon which local microstrains can be released, and it reduces the internal stress for improving the film toughness. Thermal diffusion caused by the friction heat allows Ag to diffuse and accumulate at the contact surface, and this metallic Ag of low shear stress acts as a solid lubricant. During the friction process, both graphite and Ag particles presented in the transfer layer serve as solid lubricants and thus allow the transfer layer to share its self-lubricating characteristic. The occurrence of a transfer layer prevents the Ag-DLC film from direct destructive contact with the particles. Consequently, the self-lubricating performance of the transfer layer allows the coated piston rod to have a prominent low wear rate (3.8 × 10^−9^ mm^3^/mN, Figure 5) and a longer life. This wear rate is much lower than that reported by Bin Mustafa, M.M. et al. (15–35 × 10^−9^ mm^3^/mN) [25]. By means of the transfer layer, Ag doping improves the toughness of the film. Both soft Ag particles and the graphitized top layer serve as solid lubricants and extend the service life of the friction pair.

As a result, Ag-DLC film improves the sealing performance by the interplay of three mechanisms described above.

## 5. Conclusions

A hydraulic servo-actuator is a critical aircraft control device whose sealing performance directly affects the sensitivity and accuracy of the aircraft flight attitude. This work conducted a systematical investigation of Ag-doped diamond-like carbon film to improve the seal performance. Based on the given results, the following conclusions can be drawn:(1)The vulnerable component is identified, and the service demands for the film are then proposed. The wear between the seal ring and piston rod may induce the seal issue, and the seal failure process can be divided into three stages. During the reciprocating motion of the piston rod, the particulate contaminants embedded within the seal ring make the pod surface wear. On basis of analyses of both the failure and the operating conditions, the Ag-DLC film on the piston rod is proposed to enhance its abrasion resistance, so as to improve the wear resistance of the frictional pair without affecting the initial fit clearance of the piston rod.(2)Ag-DLC films are deposited, and the film feasibility is verified by investigating the structural property and combined tribological performances. Upon increasing the Ag content, the peak corresponding to crystal face Ag (111) in the films broadens. The hardness and intrinsic stress of the films tend to decrease with the Ag content, as well as the frictional coefficient and the wear rate. The a:C-Ag_10.5%_ film exhibits optimal combined tribological performances. It is postulated that during the wear process, both the friction heat and the doped Ag promote the conversion of the sp^3^-C bonds into sp^2^-C bonds and form a graphite layer.(3)The modification mechanism behind the improvement of the sealing performance via Ag-DLC film may be a compromise of three effects. (i) The contact force and friction heat induce graphitization. (ii) The silver particles improve film toughness and act as a solid lubricant. (iii) The transfer layer plays a self-lubrication role and extends the seal life.

Our findings may offer a novel approach to make DLC film applicable for improving the seal performance of the hydraulic servo-actuator. Moreover, the present study allows us to understand the mechanism behind the applicable Ag-DLC film.

## Figures and Tables

**Figure 1 materials-13-02618-f001:**
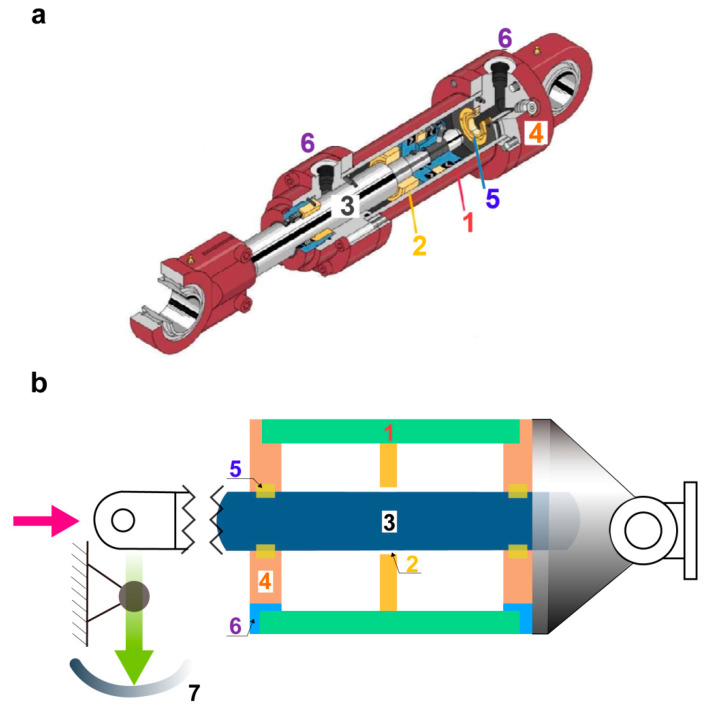
(**a**) Physical map and (**b**) Working principle of the hydraulic servo-actuator. 1. Cylinder; 2. Piston; 3. Piston rod; 4. End cap; 5. Seal ring; 6. Pipe orifice; 7. Signal feedback device.

**Figure 2 materials-13-02618-f002:**
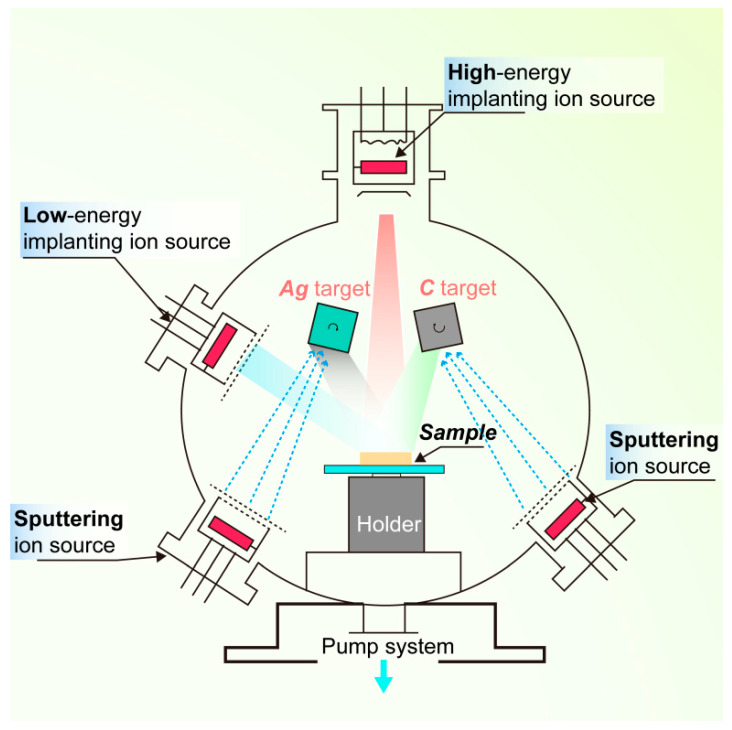
Schematic diagram of the multi-ion beam assisted deposition (IBAD) system used.

**Figure 3 materials-13-02618-f003:**
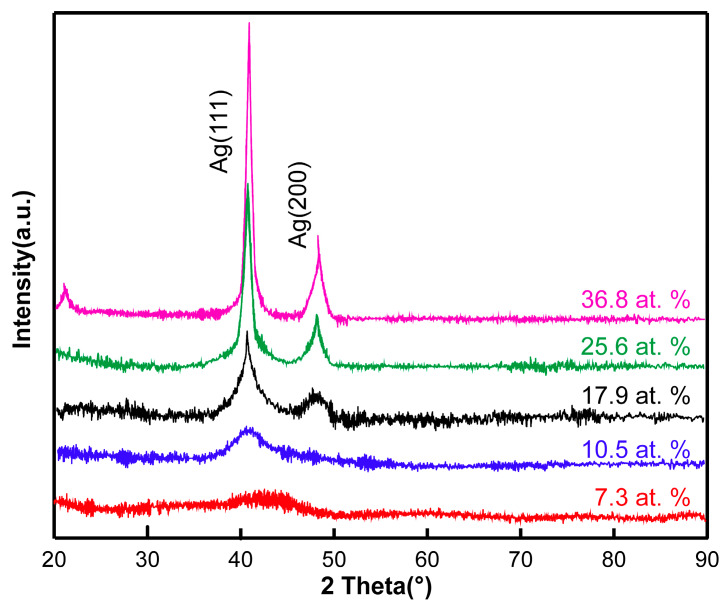
XRD diffraction patterns of Ag-doped diamond-like carbon (Ag-DLC) films with 5 Ag contents.

**Figure 4 materials-13-02618-f004:**
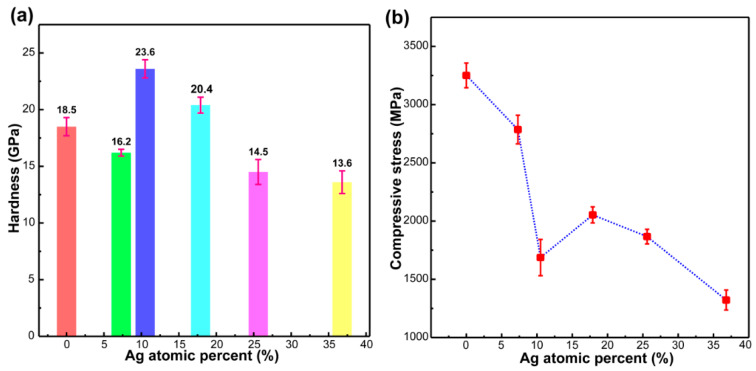
Variation of: (**a**) Hardness and (**b**) Stress of Ag-DLC films with Ag content.

**Figure 5 materials-13-02618-f005:**
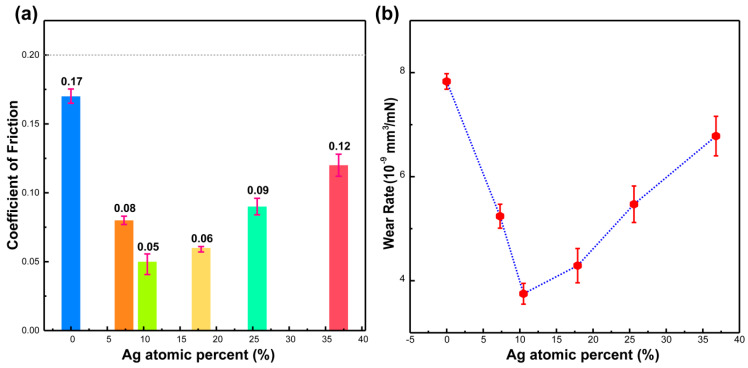
(**a**) Friction coefficient and (**b**) Wear rate of Ag-DLC films as a function of the Ag content.

**Figure 6 materials-13-02618-f006:**
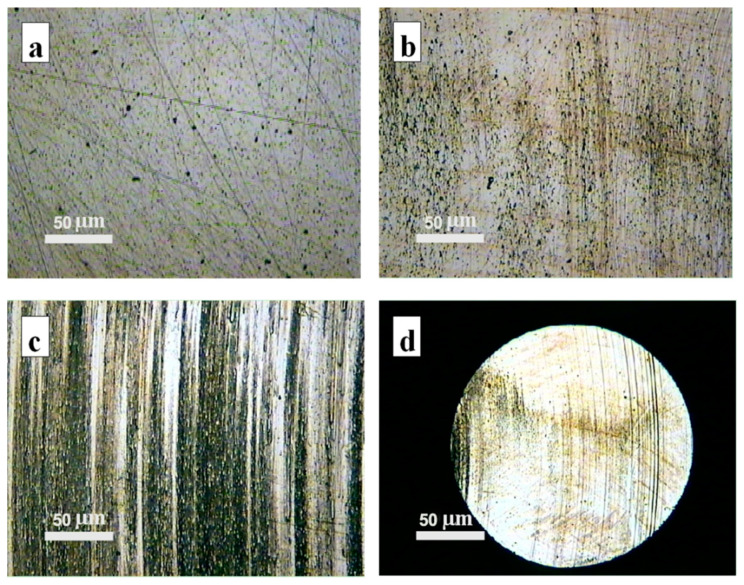
Optical microscope images of wear scars for Sample A2 during the reciprocating movement with durations of: (**a**) 15 min/coupon, (**b**) 30 min/coupon, (**c**) 60 min/coupon, and (**d**) 60 min/ball.

**Figure 7 materials-13-02618-f007:**
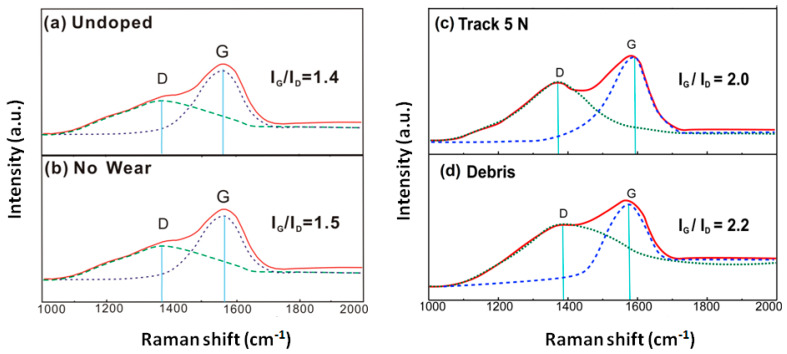
Raman spectra of the undoped DLC film and a:C-Ag_10.5%_ film. Note: (**a**) Undoped DLC film (Undoped), left top; (**b**) as-deposited film (No wear), left bottom; (**c**) wear track at 5 N load (Track 5 N), right top; and (**d**) wear debris (Debris), right bottom.

**Figure 8 materials-13-02618-f008:**
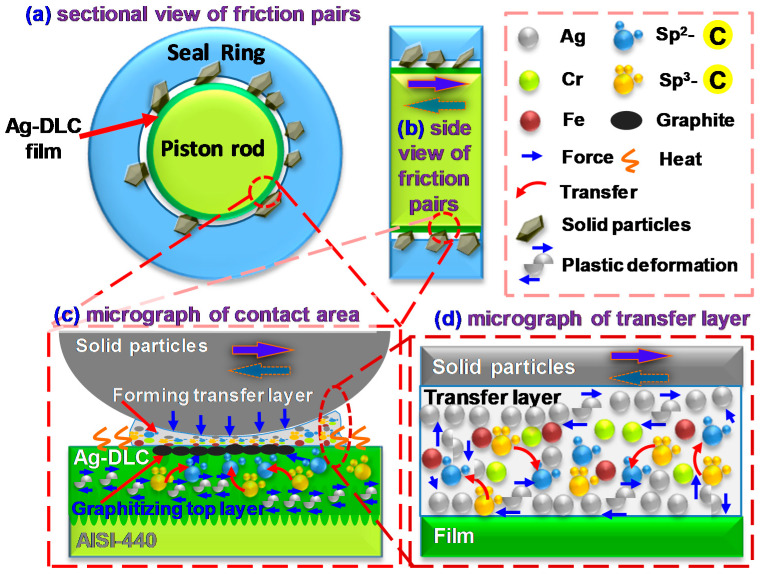
Schematic diagram of action mechanism of Ag-DLC film to improve the sealing performance. (**a**) sectional view of friction pairs, left top; (**b**) side view of friction pairs, right top; (**c**) micrograph of contact area, left bottom; (**d**) micrograph of transfer layer, right bottom.

**Table 1 materials-13-02618-t001:** Ag sputtering current and atomic percentage of six a:C-Ag_x_ film samples.

Sample	Film	Ag Sputtering Current (mA)	Ag (at.%)	Ag Grain Size (nm)
A0	a:C-Ag_0%_	0	0	0
A1	a:C-Ag_7.3%_	10	7.3	9.4
A2	a:C-Ag_10.5%_	20	10.5	14.6
A3	a:C-Ag_17.9%_	30	17.9	23.6
A4	a:C-Ag_25.6%_	40	25.6	35.3
A5	a:C-Ag_36.8%_	50	36.8	48.9

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
