# Peer review of "Attempting AG-Doped Diamond-Like Carbon Film to Improve Seal Performance of Hydraulic Servo-Actuator"

_materials, 2020, doi:10.3390/ma13112618_

Round 1

Reviewer 1 Report

MS deals with the investigation od Ag doped DLC layers using XRD, RS and mechanical tests to improve the seal performance. The text is well written and can be useful to the readers after some improvements. 

(i)   Behaviour of Ag-DLC layers is mostly governed by sp3/sp2 ratio of carbon  atoms. However, no numerical data of the ratio are provided.

(ii) RS spectrum recorded from the undoped DLC makes possible to recognize the possible changes in the ratio due to Ag doping (Fig. 7). Add, please, the spectrum.

(iii) It is generally known that Ag doping (or doping by further elements) of DLC layers induces the sp3 to sp2 transformation. Knowledge of sp3 content in the as-prepared Ag doped layers is important to reveal the proper mechanism behind the improvement of the sealing performance.

(iv) The mean size of Ag clusters also plays a significant role in assessment of the sealing performance. Add, please, the values.

Reviewer 2 Report

The paper address development of Ag-doped DLC films to improve the sealing performances in hydraulic servo-actuators in aircraft devices. The quality of the friction couple is of high importance in the good functionality and reliability of the device. The authors try to demonstrate that improved performances may be obtained by replacing W/Cr carbides in DLC coatings with softer and ductile Ag particles, enhancing the abrasion resistance of the friction pair. During analysis of this interesting paper I found some aspect that need attention to be better explained.

  1. In paragraph 3.2 film deposition raw 125-126. What do you mean by the substrate was two wafers? Please re-formulate. It seems you used 1 Si wafer for structural analysis and 1 for mechanical analysis. Why did you not used the same AISI 440 substrate for these tests? Proceeding like that, are the structural and mainly mechanical tests relevant?
  2. In paragraph 3.4. you mentioned that thickness of the substrate and films were measured. Please give their values in section 4 - Results and discussions. A SEM picture in transversal section showing these thickness values can be useful.
  3. In paragraph 3.5 please provide a more detailed description of the tribological testing system, such as: type of the test and contact, sliding distance, the testing regime (dry or wet), the formula for calculating the wear rate. In case of using lubricants (PAO oil mentioned in the paper) describe its main characteristics (composition, additives, technical specifications).
  4. What are the temperature and humidity conditions during testing?
  5. Also here it will be useful to mention the roughness of the as-deposited coated surface before starting tribological tests; if not possible, a reference or use in the paper "as-deposited". 
  6. In paragraph 4.5 a reference for the wear rates or another parameter of actual existing coatings may be useful to proof the improvement done by Ag-doped DLC coatings.
  7. Please also check and correct the punctuation of English phrases. For ex. in introduction raw 39-45 you use sometime a point (.) instead of point and comma (;). 

Reviewer 3 Report

This article is described based on many experiments carefully.
Its value is very high scientifically and industrially.

As there is a curious point, I want the authors to consider it.
At figure 7, authors estimated the IG/ID ratio of "No wear", "Track 5N",
and "Debris" with 1.5, 2.0, and 2.2, respectively.
However, I see that the intensity of the D peak in Fig. 7 increases
obviously in order of "No wear", "Track 5N", and "Debris".

In other words, the IG/ID ratio decrease in order of "No wear", "Track 5N", and "Debris".
This result becomes the important grounds of the conclusion of the authors and has the influence that is important to the later discussion.

Other points
At L.139 and L.141, the authors wrote the source materials of sputtering as only "Ag target" and "C target".
But they should write concrete material names with their purities.
At 147, the authors reported to measure the elemental composition using EDS.
I want the authors to write that any impurities except Ag and C were observed or not.

Round 2

Reviewer 1 Report

Comparison of RS of undoped and Ag doped DLC is important for the recognition of basic properties of the starting material and induced changes due to Ag doping.  

Add, please, the RS of undoped DLC in fig. 7, provide IG/ID ratio and the C sp3 values. 

Reviewer 2 Report

Dear Author,

I appreciate the completions done in the paper regarding the description of the working procedure, especially related to tribological tests. I have no further observation, the results are now clearer for the people reading the paper.
